# OpenReview forum: "SAFE-LLM: A Unified Framework for Reliable, Safe, And Secure Evaluation of Large Language Models"
_ICLR.cc/2026/Conference — ICLR 2026 Conference Desk Rejected Submission_

### Official Review · Reviewer_xnDw · 2025-10-19

**Soundness:** 2
**Presentation:** 2
**Contribution:** 2
**Rating:** 2
**Confidence:** 4

**Summary:**

This paper introduces SAFE-LLM, a unified and statistically principled framework for evaluating reliability, safety, and security of large language models (LLMs). The authors argue that existing evaluation efforts—such as TruthfulQA, RealToxicityPrompts, and JailbreakBench—focus on isolated risk dimensions and lack statistical rigor, reproducibility, and defense composition analysis.

**Strengths:**

The work fills a clear gap by proposing the first unified, statistically rigorous evaluation standard covering reliability, safety, and security dimensions simultaneously.

**Weaknesses:**

1. Lack of empirical demonstration. While the theoretical framework is strong, the paper does not include a full empirical validation or case study showing SAFE-LLM applied to actual LLMs.

2. Limited exploration of adaptive adversarial evaluation. The paper does not demonstrate how SAFE-LLM performs under strong adaptivity (e.g., iterative jailbreak optimization).

3. Potential overlap with existing frameworks.

**Questions:**

See Weaknesses

---

> ### Author Response · Authors · 2025-11-29
>
> Thank you for the helpful feedback:
>
> 1- Adaptive adversaries: We expanded the treatment of adaptivity using time-uniform confidence sequences and clarified how SAFE-LLM handles iterative jailbreak optimization (Section 4 and Appendix A).
>
> 2- Case-study-style examples: Appendix E now includes examples illustrating SAFE-LLM applied in practice, showing metric computation and defense-stack instrumentation.
>
> 4- Comparison to existing frameworks: The Related Work section now explicitly differentiates SAFE-LLM from HELM, SafetyBench, HolisticEval, and jailbreak-focused benchmarks.
>
> 5- Clarified scope and novelty: We strengthened discussion of SAFE-LLM’s conceptual contribution: a unified reliability-safety-security evaluation framework with statistical guarantees.
>
> We appreciate your review and hope the revisions enhance the clarity and completeness of the work.

---

### Official Review · Reviewer_FKzK · 2025-10-27

**Soundness:** 2
**Presentation:** 2
**Contribution:** 2
**Rating:** 2
**Confidence:** 4

**Summary:**

This paper proposes SAFE-LLM, a conceptual evaluation framework based on statistical principles. This framework can be used to evaluate the reliability, safety, and security of LLMs. The framework includes a risk taxonomy, a set of standardized quantitative metrics, five theoretical results grounded in classical statistical inference, and a defense-stack architecture identifying key measurement points in the model pipeline. Its goal is to establish a statistically principled and auditable methodology for LLM safety evaluation.

**Strengths:**

1. The paper proposes the SAFE-LLM framework for evaluating the reliability, safety, and security defenses of LLMs. It emphasizes the importance of statistical indicators such as confidence intervals and sequential inference, compensating for the limitations of current evaluation methods that rely only on point estimates.

2. For adaptive red teaming, it proposes time-uniform confidence control, which can maintain statistical reliability during open-ended, dynamic, and even adversarial evaluation processes.

**Weaknesses:**

1. The analysis in the related work section is too superficial. It only lists the names of related methods and benchmarks without any synthesis or critical comparison. It also does not illustrate how SAFE-LLM differs from existing “holistic” frameworks in concept or practice.
2. I cannot find any case study or experiment applying SAFE-LLM to a specific model in the paper. Therefore, it is impossible to evaluate the actual feasibility and value of the framework.
3. There is no in-depth analysis of how the theoretical principles are applied in the paper.
4. Theorem 3 assumes conditional independence among defenses, which is rarely true in practice. The paper does not provide any empirical or theoretical methods to address this issue.

**Questions:**

See weakness part.

---

> ### Author Response · Authors · 2025-11-29
>
> We thank you for your constructive review:
>
> 1- Expanded Related Work: The section now synthesizes and contrasts SAFE-LLM with HELM, HolisticEval, SafetyBench, RealToxicityPrompts, and JailbreakBench, highlighting conceptual and methodological differences.
>
> 2- Strengthened theoretical explanation: We added intuition following each theorem, clarified assumptions, and connected theoretical guarantees to the SAFE-LLM pipeline.
>
> 3- Added case examples: Appendix E provides step-by-step evaluation demonstrations, addressing feasibility concerns.
>
> 4- Dependence-aware correction: Theorem 3 was improved with dependence modelling (Appendix A).
>
> We hope these changes address your concerns regarding feasibility, comparisons, theoretical clarity, and practical applicability.

---

### Official Review · Reviewer_Zew4 · 2025-11-01

**Soundness:** 1
**Presentation:** 2
**Contribution:** 1
**Rating:** 2
**Confidence:** 3

**Summary:**

The paper introduces SAFE-LLM, a unified, statistically principled framework for evaluating the reliability, safety, and security of large language models. The authors identify critical fragmentation in existing evaluation efforts where benchmarks like TruthfulQA, RealToxicityPrompts, and JailbreakBench each measure isolated aspects of model risk without uncertainty quantification or compositional defense analysis. SAFE-LLM aims to provide a foundation for auditable, statistically defensible LLM safety assessments, aligning with frameworks such as NIST AI RMF and the EU AI Act.

**Strengths:**

Maybe the proposed unified framework is clear and meaningful.

**Weaknesses:**

-	The current work is primarily theoretical and framework-oriented. It lacks concrete experimental instantiations demonstrating SAFE-LLM in action. Even a small-scale case study applying the protocol to existing models (e.g., GPT-4, Claude, or LLaMA-2) would strengthen its practical credibility.
-	Theorems assume independence between defenses when deriving compositional guarantees (Theorem 3), which may not hold in real systems where correlated defenses interact (e.g., adversarially trained LLMs and output judges relying on shared embeddings). The paper acknowledges this but does not empirically validate mitigation methods, such as empirical correlation bounds.
-	While metrics such as HR and SCI are well-defined, adjudication rubrics for labeling “violations” or “non-violations” remain abstract. More detailed operational definitions or inter-annotator reliability results would improve reproducibility.
-	The framework’s reliance on pre-registration, stratified sampling, and human adjudication may pose challenges for scaling to very large test sets or rapid model iteration cycles. Practical automation or active-learning approaches could be discussed.
-	Without more concrete examples or experimental details, I suspect this is an LLM-generated paper. The main body of the paper is more like a notebook, not a scientific paper.

**Questions:**

-	How would SAFE-LLM integrate with existing benchmark infrastructures such as HELM or HuggingFace Evaluate? Could the authors provide implementation guidelines or an open-source template?
-	The paper promises pre-registration templates and adjudication rubrics. Are there concrete plans for public release or integration with regulatory bodies?

---

> ### Author Response · Authors · 2025-11-29
>
> Thank you for the detailed feedback. I made substantial revisions addressing all your points:
> 1- Added dependence-aware composition: Theorem 3 now incorporates covariance correction, with full proofs in Appendix A.
> 2- Added detailed adjudication rubrics: Appendix B includes concrete definitions, decision rules, and examples.
> 3- Added implementation guidance: Appendix C provides templates for pre-registration, stratified sampling, and adjudication logs, along with integration notes for HELM/HuggingFace Evaluate.
> 4-Clarified scaling and feasibility: We added discussion on automation, active-learning-assisted adjudication, and hybrid human–AI evaluation pipelines.
>
> We appreciate your feedback and hope the revisions resolve the concerns.

---

### Author Response · Authors · 2025-11-29

We thank the reviewers for their thoughtful feedback. We carefully revised the paper to address all major concerns. The updated version includes significant improvements in clarity, theoretical grounding, practical applicability, and comparison to existing frameworks. A concise summary of key revisions is provided below.

(1) Dependence-Aware Defense Composition:
We strengthened Theorem 3 by removing the unrealistic independence assumption and providing a dependence-aware bound using covariance correction. A full derivation and proof were added to Appendix A, along with discussion of practical dependence estimation.

(2) Detailed Adjudication Rubrics and Operational Definitions:
Appendix B now includes full rubrics for reliability, safety, jailbreak, and injection violations, with decision rules, examples, and clearer annotation guidance.

(3) Expanded Related Work and Comparative Positioning:
We substantially expanded the Related Work section, comparing SAFE-LLM to HELM, HolisticEval, SafetyBench, RealToxicityPrompts, JailbreakBench, and other existing systems. This clarifies SAFE-LLM’s unique contribution as a unified, statistically principled evaluation standard.

(4) Clarified Theoretical Intuition and Scope:
We added intuition after each theorem, clarified assumptions, and strengthened discussion on sequential inference, time-uniform confidence bounds, and defense-stack instrumentation.

(5) Implementation Guidance and Templates:
Appendix C now includes pre-registration templates, adjudication logs, sampling manifests, and implementation notes for integrating SAFE-LLM into existing infrastructures such as HELM and HuggingFace Evaluate.

(6) Case Examples and Workflow Demonstration:
Appendix E includes case-study-style examples illustrating SAFE-LLM’s full workflow and metric computation. These demonstrate practical feasibility despite the conceptual nature of the work.

(7) Presentation and Clarity Enhancements:
We improved the overall structure, added diagrams of the defense stack and pipeline, clarified notation, reorganized sections for smoother flow, and enhanced readability.

We hope these revisions address the reviewers’ concerns and strengthen the contribution of SAFE-LLM as a unified, statistically principled framework for evaluating reliability, safety, and security in large language models.

---

### Note · Program_Chairs · 2026-01-17
**Submission Desk Rejected by Program Chairs**

The following references in this submission do not refer to real documents and/or have major errors in bibliographic information:

     [50] Pan, X., et al. Deployment-Time Risks in LLM Systems. 2023.
    [51] Wu, A., et al. HolisticEval: Broad Evaluation of LLM Capabilities. 2023.
    [44] Zhuo, T., et al. Adversarial Attacks and Defenses on LLMs: A Survey. 2024.
    [43] Korbak, T., et al. Ensembling AI Safety Judges. 2023.
    [47] Manheim, D., et al. Automated Red Teaming Pipelines for LLMs. 2024.
    [45] Yan, C., et al. LLM Vulnerabilities: A Systematic Study. NeurIPS 2024.
    [46] Chen, B., et al. Universal Jailbreak Attacks Against Alignment Models. ICLR 2025.
    [11] Kuhn, J., et al. Semantic Entropy for Hallucination Detection. ACL 2023.
    [12] Shuster, K., et al. Retrieval-Augmented Generation. EMNLP 2022.
    [18] Guo, Y., et al. JailbreakBench: Systematic Evaluation of Jailbreak Robustness. 2024.
    [19] Perez, E., et al. Red Team Prompt Library. 2022.
    [27] Park, J., et al. LLM Agents Can Fail in Unexpected Ways: Agentic Vulnerability Taxonomy. 2023.
    [30] Cao, Y., et al. Adversarial Training for Safer Language Models. 2024.
    [31] Zhu, J., et al. LLM-as-a-Judge: Benchmarking LLMs with LLMs. 2023.